# Development of Simvastatin-Loaded Particles Using Spray Drying Method for Ex Tempore Preparation of Cartridges for 2D Printing Technology

**DOI:** 10.3390/pharmaceutics15092221

**Published:** 2023-08-28

**Authors:** Barbara Sterle Zorec, Rok Dreu

**Affiliations:** Faculty of Pharmacy, University of Ljubljana, Aškerčeva Cesta 7, SI-1000 Ljubljana, Slovenia; rok.dreu@ffa.uni-lj.si

**Keywords:** inkjet printing, spray drying, design of experiments, simvastatin, antioxidants, stability, ink medium, nanocellulose, particle size, emulsion

## Abstract

In this work, a spray drying method was developed to produce drug/polymer (simvastatin/polycaprolactone) microparticles that have the potential to be used as a pre-formulation for ex tempore preparation of 2D printing cartridges. An experimental model was designed with the process parameters set to predict the smallest particle size required for successful 2D printing. Three different types of particles (lactose, nanocellulose/lactose, calcium silicate) were produced, and the average size of the dry particles varied depending on the sampling location (cyclone, collection vessel). The encapsulation efficiency of simvastatin was highest with nanocellulose/lactose from the collection vessel. The one-month stability of simvastatin in the particles showed low content, but the addition of ascorbic acid as an antioxidant increased the chemical stability of the drug. Interestingly, the addition of antioxidants decreased the stability of simvastatin in the calcium silicate particles from the collection vessel. Dispersion of the particles in three different propylene glycol and water mixtures (10/90, 50/50, and 90/10% (*v*/*v*)), representing a printable ink medium with three different viscosity and surface tension properties, showed that nanocellulose/lactose was the most suitable antiadhesive in terms of dispersed particle size (˂1 µm). After one month of storage, the dispersed particles remained in the same size range without undesirable particle agglomeration.

## 1. Introduction

The current health care system is insufficient in meeting the needs of all individuals, as commercially available medicines are offered only in a few limited doses. This is particularly problematic for drugs with a narrow therapeutic window, highly potent drugs, and formulations intended for children or the elderly [1,2]. The concept of personalized medicine should extend beyond age or physiological conditions to include the metabolic state and pharmacogenomics of the individual [3,4], thereby improving treatment efficacy and patient compliance [5]. 

The widespread application of the personalized approach is highly dependent on the development of rapid and adaptable technologies that enable the production of individualized drug delivery systems [6]. Among other technologies, two-dimensional (2D) printing technology has been proposed. 2D printing enables the precise deposition of drugs dissolved or dispersed in an ink base medium onto various edible substrates, resulting primarily in solid oral dosage forms. There are several 2D printing technologies, yet piezoelectric and thermal inkjet printing are the most widely studied as they are known to be fast and flexible compared to others [6]. One of the advantages of inkjet printing technology is also the ability to produce on-demand drug delivery systems even at the point of dispensing, such as a pharmacy or hospital, using non-destructive spectroscopic techniques (e.g., near-infrared (NIR) spectroscopy) to verify the drug assay onsite [7,8,9,10]. 

Various personalized drug-containing dosage forms have been reported to be produced using inkjet printing technology [11,12,13,14,15,16], including mucoadhesive buccal films [17] as well as direct printing on the nails [18] and even on contact lenses [19]. 

With regard to ink formulations, numerous previous studies have mainly focused on printing drugs from drug solutions (water- or non-water-based) [14], but also from drug-loaded nano/microparticle dispersions [20]. Nano/microencapsulation of the drug into nano/microparticles (e.g., polymeric [21,22,23], inorganic [24], lipid [25]) can successfully overcome the potential limitations of low stability, solubility, and bioavailability of drugs [26]. However, regardless of the type of formulation used (solutions or particles dispersion), several physicochemical parameters must be considered for the drug-containing inks to be printable. The most important of these are viscosity, density, surface tension, and, in the case of particle dispersion, also particle size [27]. Still, these parameters depend on the characteristics of the drug and the specifications of the printer used [14]. In addition to the above stated parameters, the physical stability of the ink formulations and the chemical stability of the drug in the ink base medium must be ensured throughout the storage and use period if we want this type of technology to be more widely commercialized and available to patients.

However, in the case of nano/microparticle dispersions as ink formulations, their physical instability can pose a major challenge in terms of printability and non-uniform drug delivery. If the distribution of drug-containing particles in the ink is non-uniform, this also applies to the final product after the printing process, resulting in non-uniform dosage forms. In addition, potential particle agglomerates in the ink can cause nozzle clogging during the printing process and affect the efficiency of the process [20].

These challenges can be faced by using techniques providing nearly monodispersed particles in size and shape and with that diminishing the possibility of particle agglomeration and consequently of instability of such systems. One of the suitable methods for the preparation of polymeric nano/microparticles is spray drying, which has been widely used in pharmaceutical research for decades and can be used to obtain nearly monodisperse particles suitable for the purpose [28].

The aim of this study is to propose a solution to the problem of physical and chemical instability of the final ink formulations. The proposed approach is to prepare the ink formulation ex tempore immediately before its use by simply adding the pre-prepared dry drug-containing particles to the ink base medium, resulting in a particle dispersion suitable for 2D inkjet printing. This “extemporaneous compounding” approach shortens the required shelf life of the classical drug-containing ink formulation, but at the same time, extends the shelf life of the dry particle pre-formulation.

Namely, a method for preparing particles by spray drying organic emulsions containing a model drug simvastatin, a polymer, and various antiadhesive substances was developed. The dry particles produced were tailored by Design of Experiments (DoE) to exhibit the appropriate physical properties for successful 2D printing after being redispersed in a blank ink medium. After particle fabrication, the chemical stability of the drug in the dry particles was tested with and without the addition of antioxidants in the initial formulation to evaluate the potential longer stability of simvastatin in the particles. The prepared dry particles were then redispersed in a blank ink medium with different volume ratios of propylene glycol (PG) and water, and finally, the one-month physical stability of these redispersed particles was evaluated.

## 2. Materials and Methods

### 2.1. Materials 

Simvastatin (SIM) was kindly donated by Krka d.d. (Novo Mesto, Slovenia); polycaprolactone (PCL) of 14,000 daltons, polysorbate (Tween ^®^ 20), propylene glycol, and ascorbic acid (AA) were purchased from Sigma Aldrich, St. Louis, MO, USA. Lactose Mesh 200 donated by Lek d.d. (Slovenia), Nanocelullose (NCC, Cellu Force, Montreal, QC, Canada), and Florite R calcium silicate (CaSi, Tomita Pharmaceutical, Tokushima, Japan) were used as antiadhesives in spray drying experiments. Chloroform (Merck KGaA, Darmstadt, Germany) and purified water were used as O/W emulsion phases. The solvents used for U(H)PLC analysis were of HPLC grade. All other reagents used were of analytical grade. Water for UPLC analysis was purified using a Milli-Q system with a 0.22 Millipak 40 filter (Millipore, Cork, Ireland). 

### 2.2. Methods

#### 2.2.1. Preparation and Characterization of Emulsion

The emulsions were optimized by using different amounts and types of surfactants as well as the ratios between the inner and outer phases and adjusted accordingly. The four emulsions listed in Table 1 (EMU 1, EMU 2, EMU 3, and EMU 4) consist of an organic phase (0.8 g SIM, 1.6 g PCL in chloroform) and a water phase (4 g Tween^®^ 20 in water), with the most optimal emulsion being EMU 2, which contains 40 g chloroform and 120 g water. In a case of formulation with antioxidant, 0.2% (*w*/*w*) of AA, relative to the total weight of the dry components, was added in the water phase of the emulsion. Emulsions were prepared using an Ultra-Turrax mixer (IKA T25 digital, IKA, Staufen, Germany) at 8000 rpm for 5 min and at 12,000 rpm for 3 min and were also homogenized using an APV 2000 high-pressure homogenizer (SPX Flow, Charlotte, NC, USA) with three passes at 300 bars.

The droplet size and polydispersity index (PDI) of the organic phase of the prepared O/W emulsion were measured and evaluated using a Zetasizer (Nanoseries-ZS, Malvern Panalytical, Malvern WR14 1XZ, UK). All samples were prepared in triplicate and expressed as average ± standard deviation (SD).

#### 2.2.2. Particle Preparation by Spray Drying Emulsions

A spray dryer (Mini Spray Dryer B-290, Büchi, Flawil, Switzerland) was used to produce dry particles from initial O/W emulsions. A two-fluid nozzle with an orifice diameter of 1.4 mm and an orifice diameter of the cap of 2.20 mm was used. The parameters of the spray drying process were as follows (depending on the setting of the DoE experimental design): the flow rate of the drying gas varied between 20–38 m3/h (50–100% aspiration rate); the inlet temperature was kept between 110 and 170 °C; the spraying rate was 0.3–3.9 mL/min (1–13% of the maximum rotation of the peristaltic pump); the flow meter spraying air (atomization gas flow rate) was varied between 40 and 60 mm (473 and 742 L/h); and the nozzle geometry was varied between the nozzle and cap being parallel, the cap being half-screwed and fully screwed (0–0.75 mm). The nozzle geometry setting is defined as 0 mm when the nozzle tip is parallel to the nozzle cap and 0.75 mm when the nozzle cap is fully screwed on, as previously shown by Pohlen et al. [29]. After completion of the spraying process, the dry particles were removed from the cyclone and the product collection vessel and stored separately, with two samples per experiment.

Antiadhesive substances. Before spray drying, 8% (*m*/*v*) or 2.5% (*w*/*v*) of different antiadhesives were added to the final emulsion formulation. Lactose (8% (*w*/*v*)), a combination of NCC and lactose (25:75% (*w*/*w*); 8% (*w*/*v*)), and CaSi (2.5% (*w*/*v*)) were selected as optimal. 

#### 2.2.3. Experimental Design

Initially, emulsion formulations consisting of lactose, Tween^®^ 20 (water phase) and PCL, and simvastatin in chloroform (organic phase) were tested by spray drying to determine the limits beyond which the formulations could not be processed. Minitab^®^ 17 software (Minitab Inc., State College, PA, USA) was used for experimental design and statistical analysis. A response surface design with five process variables was used, and four replicates were performed at the central point to estimate the process replicate error. The five independent variables were inlet temperature (X1), drying gas flow (X2), spray rate (X3), nozzle pressure (X4), and nozzle geometry setting (X5). The mean particle size, d50 particles (Y), was used as the DoE response for optimization. A total of 46 experiments were performed with four replicates in the central point, as shown in Table 2. Stepwise elimination with criterion α = 0.15 was used in setting the models to eliminate variables that were not significant for a given response. Once the models were established, 2 optimization experiments were performed to optimize the selected DoE responses based on the location of the collected samples (either from the cyclone or from the sample collector). The Minitab^®^ Response Optimizer was used for the local minimum/maximum, depending on the desired characteristic of the product. Each model was evaluated based on the three coefficients of determination—R^2^, R^2^(adjusted), and R^2^(predicted)—and based on the root mean square mean error (RMSE) and the normalized RMSE (NRMSE). The latter was normalized by dividing the RMSE by the average of the observed values.

#### 2.2.4. Characterization of Prepared Particles 

Particle Size Analysis. Dry particle size distributions were measured by a laser diffraction measurement (Mastersizer S, Malvern Instruments, Ltd., Malvern WR14 1XZ, UK) using the 300 F lens and a dry powder feeder unit with the following parameters: measuring range from 0.5 to 880 µm, density of 0.8 g/cm^3^, feed air pressure 5 bar, 0.5–5% obscuration rate, and Fraunhofer theory setting. As for the droplet size, the particle size distribution was described by volume-based distribution parameters d10, d50, d90, and SPAN, where SPAN was calculated as SPAN = (d90 − d10)/d50. Measurements were undertaken in triplicate and expressed as average ± SD. 

Scanning Electron Microscopy. Scanning electron microscopy (SEM) was used to determine the morphology of the spray-dried particles. The dry particles were placed on a carbon tape and examined using a Supra 35 VP, high-resolution scanning electron microscope (Carl Zeiss, Oberkochen, Germany) at 1.0 kV acceleration voltage and different magnifications using secondary detector.

Moisture Content. The moisture content of the dry particles was determined gravimetrically as loss on drying, utilizing the Büchi moisture analyzer (B-302, Büchi, Switzerland) by heating approx. 3 g of powder for 15 min at 85 °C. The moisture content was calculated using Equation (1).
(1)$$Moisture\;content\;\left(\%\right)=\frac{sample\;mass\;\left(before\;drying\right)-sample\;mass\;(after\;drying)}{dry\;particle\;mass}*100$$

#### 2.2.5. Drug Loading and Encapsulation Efficiency 

Drug encapsulation efficacy experiments were performed to determine the actual content of simvastatin entrapped in the particles. Drug loading (DL) and encapsulation efficiency (EE) were calculated according to the Equations (2) and (3), respectively. 

Samples for UPLC were prepared differently depending on the antiadhesives used. In the case of lactose and NCC/lactose, 120 mg of the dry particles were first dissolved in 5 mL of water (to dissolve the antiadhesives) and centrifuged at 3000 rpm for 10 min, and then, the supernatant was removed. The sediment was dissolved in 2 mL of chloroform and diluted to 50 mL with methanol. The prepared solution was further diluted with methanol, accordingly, corresponding to approximately 25 µg/mL of simvastatin and analyzed with UPLC (Section 2.2.7). Silicate samples for UPLC were prepared by first dissolving the particles in 2 mL chloroform and further diluting with methanol accordingly. All samples were prepared in triplicate and expressed as average ± SD.
(2)$$DL\left(\%\right)=\frac{determined\;mass\;of\;drug\;in\;particles}{mass\;of\;drug-loaded\;particles}*100\%$$
(3)$$EE\left(\%\right)=\frac{determined\;mass\;of\;drug\;in\;particles}{theoretical\;mass\;of\;drug\;in\;particles}*100\%$$

#### 2.2.6. Stability Study

The drug chemical stability of the dry particles was evaluated at room (25 °C, 40% RH) and accelerated conditions (40 °C, 75% RH). A known amount of the prepared drug-containing particles with and without AA was stored in the exicator and stability chamber for 1 month. After this period, the particles were analyzed as described in Section 2.2.9. The degradation of simvastatin in the particles was calculated as the relative content of simvastatin after 1 month according to the Equation (4). All samples were prepared in triplicate and expressed as average ± SD.
(4)$$Drugcontent=\frac{drug\;content\;\left(t=1\;month\right)}{drug\;content\;\left(t=0\right)}*100\%$$

#### 2.2.7. Preparation and Characterization of Blank Ink Media

Three blank ink media with different PG:water volume ratios, namely 10/90, 50/50, and 90/10% (*v*/*v*) were prepared. 

Viscosity measurements. The viscosity of three different blank ink media was measured using a dual-spaced measuring system (DIN 54453) on an Anton Paar rheometer, Physica MCR301. Viscosity was determined at 25 °C and a shear rate ranging from 90 1/s to 100 1/s. Measurements were undertaken in triplicate and expressed as average ± SD.

Surface tension measurements. The surface tension of PG:water mixtures described above was measured at 25 °C and determined with a K12 tensiometer (Kruss GmbH, Hamburg, Germany) using a Wilhelm plate. Measurements were undertaken in triplicate and expressed as average ± SD.

#### 2.2.8. Redispersion of the Dry Particles

Three different types of prepared dry particles (containing lactose, NCC/lactose (25:75; %(*w*/*w*)), and CaSi as antiadhesives) were dispersed in three different PG and water mixtures (10/90, 50/50, and 90/10% (*v*/*v*)), using two different mixing procedures. An amount of 100 mg of the prepared particles were weighed in a test tube by adding 20 mL of the selected blank ink medium. The samples were mixed manually and/or vortexed for an additional 2 min. The measurements were performed in triplicates and expressed as average ± SD. 

Redispersed particle size measurements. The size of redispersed particles was measured by a laser diffraction measurement (Mastersizer S, Malvern Instruments, Ltd., UK), using a small volume dispersion unit, a 300RF lens with a measurement range from 0.05 μm to 880 μm, set a polydisperse analysis model, a density of 0.8 g/cm^3^ and entered the refractive indexes for each mixture separately. The particles dispersed in the PG /water mixture were added to purified water until the level of shading of the incoming laser light reached at least 2%, and the size of the dispersed particles was measured in triplicate.

#### 2.2.9. UPLC Analysis

The UPLC method was developed and described previously [30]. Simvastatin was determined using the Acquity UPLC chromatographic system (Waters Corp., Milford, MA, USA). A UV-VIS photodiode array (PDA) module equipped with a high-sensitivity flow cell was used for detection. The column used was an Acquity UPLC BEH C18 1.7 μm, 2.1 × 100 mm reversed-phase column (Waters Corp., USA). Gradient elution was used with mobile phase A (90% water, with 0.1% orthophosphoric acid and 10% acetonitrile) and mobile phase B (98% acetonitrile, 2% water). The gradient method was as follows: start at 50:50 (A:B); 0–6 min, 50:50–40:60; 6–7 min, 40:60; 7–8 min, 40:60–50:50; 8–10 min, 50:50. The flow rate was set at 0.3 mL/min and the column temperature was maintained at 45 °C. The autosampler temperature was set at 10 °C. The injection volume was 5 μL, and the run time was 10 min. Simvastatin and its acid form were detected at a wavelength of 238 nm and retention times of 4 min and 6 min, respectively. The standards for the calibration curve were prepared in methanol covering the range of 0.5–30 µg/mL with R2 0.9996.

#### 2.2.10. Statistical Analysis

Statistical analysis was performed with Minitab^®^ 17 software (Minitab Inc., PA, USA), using Tukey’s test (one-tailed ANOVA) and paired t-tests, with significance considered to be 0.05. Data were expressed as average ± standard deviation (n = 3).

## 3. Results and Discussion

### 3.1. Emulsion Optimization

Simvastatin is a well-known active pharmaceutical ingredient (API) that is poorly soluble not only in aqueous medium, but also in the vast majority of organic solvents, which also applies for PCL [31,32,33]. Therefore, we were quite limited in choosing an organic solvent in which both PCL and simvastatin possessed good solubility. Among some others, chloroform proved to be an optimal choice both as a solvent and as an organic phase in the emulsion.

After preliminary optimization of the emulsion, the emulsion described in Section 2.2.1. was examined for different ratios between the inner and outer phase (Table 1). The results show that the lower the water content, the larger the droplet size and polydispersity index (PDI) of the emulsions (Table 1). Since the emulsion with a water to chloroform ratio of 3:1 had an adequate distribution of droplet size and sufficient stability for at least 24 h, EMU 2 was selected for further experiments. The final emulsion used as formulation for spray drying contained 120 g water, 40 g chloroform (EMU 2), and 20% dry matter: consisting of simvastatin, PCL, Tween^®^ 20 as described in Section 2.2.1.

### 3.2. The Size of the Dry Particles after Spray Drying 

The particle size is an important feature in terms of the printability of the system in which the prepared particles are dispersed in a certain blank ink medium [34]. The particle size should be as small as possible to avoid clogging the print nozzle, although the acceptable size range depends on the type and printhead characteristics of the printer used. Pardeike et al. report that when using a piezoelectric microdrop MD-K-140 printer, the particle size should not exceed 5 µm to be printable without clogging the nozzle [20] while others reported that the particles should generally be 20 times smaller than the diameter of the nozzle [14]. However, since the dry particles are expected to be further dispersed in a blank ink medium, which should contain the particles of a certain size, we are mainly interested in the size and behavior of the particles after their redispersion. Since we expect the redispersed particles to be in a comparable size range to the same dry particles before redispersion, we have optimized our spray drying process to obtain the smallest dry particles possible. Prior to spray drying, lactose was added to the final emulsion as an antiadhesive substance. 

The size of dry particles produced by different parameters of the spray drying process of our emulsion varied from 5 μm to 165 μm. The optimal parameters for spray drying were selected based on the experiments proposed by DoE using the process parameters as factors (Table 2). We also separately measured the size of particles collected from the collection vessel and particles adhering to the cyclone walls and found that particle size varied depending on the collection location. This is consistent with a previous study by Lindeløv and Wahlberg, which was conducted with a slightly different spray drying configuration and for a different purpose [35]. 

Later, we combined the samples from the cyclone and the collection vessel and measured the size again. Our d50 results, shown in Table 2, indicate that the particle size distributions of the combined samples are mainly determined by the particles collected at the cyclone walls. Therefore, we processed the results of the collection vessel and the combined samples separately to obtain two different optimization protocols for the parameters of the spray drying process from two different models set.

First, based on the results obtained from the collection vessel only (CV), a quadratic response surface model was fitted, deriving the following equation:$$\begin{array}{c} CV=993-2.453\times X1-18.37\times X2+23.76\times X3-2.755\times X4+0.0976\times {X2}^{2}\\ \;\;\;\;\;\;\;\;\;\;\;\;+0.02867\times X1\times X2-0.1280\times X1\times X3-0.2339\times X2\times X3+0.316\times X3\times X4 \end{array}$$
with relatively high coefficient of determination: R^2^ = 0.8185, R^2^(adj) = 0.7731, R^2^(pred) = 0.6716. We also calculated RMSE and NRMSE between the observed values and those predicted by the model. The RMSE is 32.61 µm, and the NRMSE is 56.93%, indicating a low quality of the model.

However, the optimized parameters of the spray drying process (Table 2, E47) resulted in sufficiently small particles, regardless of the sampling location.

Then, the second model was established, considering only the size measurements obtained from the combined product (CP). Compared to the previous model, the following equation:$$\begin{array}{l} CP=104-0.092\times X1+2.625\times X2-10.70\times X3-3.67\times X4+0.722\times {X3}^{2}\\ \;\;\;\;\;-0.00882\times X1\times X2-0.0471\times X1\times X3+0.0179\times X1\times X4-0.0214\times X2\times X4\\ \;\;\;\;\;+0.1112\times X3\times X4\; \end{array}$$
resulted in a slightly lower coefficient of determination (R^2^ = 0.7098, R^2^(adj) = 0.6296, R^2^(pred) = 0.4265). Nevertheless, the quality of the model was significantly improved, as the RMSE between the observed and the model predicted values is 17.15 µm, although it is still not the best possible one, as the NRMSE is slightly higher compared to the previous model (58.12%). However, based on the model for the combined product, optimized process parameters were established (E48), resulting in the smallest particles of all experiments and confirming our established model (Table 2).

The model shows that four out of five process parameters tested have an important influence on the size of the generated particles, namely inlet temperature, drying gas flow, nozzle pressure, and spraying rate, with the exception of nozzle geometry (the *p*-value in the model is too high and shows no significance).

A look at the surface plots shows that temperature in conjunction with nozzle pressure has no noticeable effect on particle size (Figure 1b). However, when including the spraying rate, the model suggests using mainly the highest inlet temperatures (Figure 1c), which also corresponds to the relationship with the drying gas flow (when it has its highest value) (Figure 1a). In accordance with this suggestion, we used both the highest temperature and the highest drying gas flow in our optimized protocol to produce the smallest particles. 

Nevertheless, a higher temperature should primarily lead to a higher Peclet number and thus larger particles [36,37]. However, the Peclet number can be manipulated by the evaporation rate of the chosen solvent and the diffusional coefficients of the solutes we use in the formulation [28], which may lead to a different result that is in better agreement with our result [28,38,39].

In correlation with other process variables, our model suggests keeping the spraying rate at the middle values to obtain the smallest particles (Figure 1c,e,f). However, increasing the spraying rate leads to a decrease in mass transfer rate during spray drying and a decrease in droplet size and final dry particles [38]. However, in some cases, droplet collisions and coalescence can lead to the opposite phenomenon, which was also seen in our results [40].

Moreover, the highest nozzle pressure results in smaller particles, regardless of the values of the other parameters (Figure 1b,d,f). This result was expected since the droplets formed during atomization also determine the size of the resulting dry particles. The higher the nozzle pressure, the smaller the droplets, both of which lead to a smaller size of the dry particles [41,42].

Using the optimized protocol provided by the CP model, we can see that the smallest particles were mostly produced primarily at higher inlet temperatures, higher or medium drying gas flow settings, higher nozzle pressures, and medium spraying rates.

Since the specifications for inkjet printing were aimed at obtaining the smallest particles possible, the E48 protocol proposed by the second model, which comes only from CP, was used for further experiments.

### 3.3. Antiadhesive Substances

The addition of an antiadhesive to the final emulsion formulation was necessary for our emulsion to be processed by the spray drying method. Since PCL melts at low temperature (56–64 °C) [43], it is difficult to use higher temperatures to prepare the particles [44]. For this reason, different antiadhesives, namely lactose, NCC/lactose, and CaSi, were added to the outer water phase of the emulsion and with that different particles were produced. 

Moreover, the selected antiadhesives have very different physiochemical properties, and thus, we adjusted the method of formulation preparation to obtain a stable emulsion containing dissolved or suspended antiadhesives. In the case of lactose and NCC/lactose, a colloidal solution was formed, while in the case of silicate, due to its low solubility in the aqueous phase, a suspension of silicate was prepared in the outer water phase of the emulsion.

As a result, after spray drying the different formulations, three different types of dry particles were produced, which exhibited different physical properties, especially in terms of size and morphology (Figure 2, Table 3). 

In Figure 2, we see that the lactose and NCC/lactose particles are composed of smaller particles, forming much larger clusters (Figure 2a,b), while the particles containing CaSi are present as large independent structures (Figure 2c). However, the median size measured by laser diffraction of the dry particles was 5.36 µm for the lactose, 5.47 µm for the NCC/lactose, and 14.81 µm for the CaSi particles (Table 3). We see the discrepancy between the laser diffraction measurements and the sizes on the images from SEM. Namely, the lactose and the NCC/lactose clusters are much larger than the measured 5 µm. Apparently, using 5 bar pressure to measure the size of the dry particles by laser diffraction broke the clusters into smaller particles, which were then measured.

We again compared the size of the particles collected separately (cyclone and collection vessel) and found that the particles collected from the collection vessel were generally slightly smaller than those collected from the cyclone walls, regardless of the antiadhesive used. This result was to be expected since the larger particles are more likely to stick to the cyclone walls, as previously shown [45]. 

### 3.4. Drug Loading and Encapsulation Efficiency of Simvastatin in the Dry Particles

Encapsulation efficiency is very important in the conversion of emulsions into solid dry particles because a considerable amount of the active ingredient may be lost during the spray drying process due to the chemical instability of the drug, decreasing the amount of encapsulated active ingredient and increasing the production cost [46,47].

In the case of 2D printing, it can be quite challenging to print a sufficient amount of the drug on the edible substrate. Therefore, to produce a 2D-printed drug delivery system with a higher drug concentration, the concentration of the drug in the ink should also be correspondingly higher. For this reason, the use of higher concentrations of the drug in the initial formulation is necessary. 

Figure 3 shows that the encapsulation efficiency immediately after particle preparation is lowest for lactose in the cyclone (36.6%) and highest for NCC/lactose particles (88.3%) from the collection vessel. However, even for CaSi, drug encapsulation exceeds 70% regardless of sampling location (72.9% in cyclone, 80.3% in collection vessel). The lower encapsulation efficiency of all three particle types deposited on the cyclone walls could possibly be due to the higher temperature we observed in the cyclone compared to the collection vessel, which could affect the stability of the drug during the spray drying process.

In addition, according to the relevant literature, simvastatin is known to be an oxidatively and hydrolytically unstable molecule. For this reason, antioxidants have been used as stabilizers in various formulations in the past to prevent the oxidation of simvastatin [30,48,49,50]. Therefore, AA was added to the initial emulsion formulation to potentially preserve chemical stability of simvastatin during the spray drying process.

The addition of AA significantly increased the encapsulation efficiency for the lactose particles from the collection vessel and for the NCC/lactose particles from the cyclone. The change was insignificant for the lactose and NCC/lactose particles sampled elsewhere. Surprisingly, the encapsulation efficiency of the CaSi particles decreased significantly when AA was added to the formulation before spray drying, as shown by our results.

Simvastatin final products on the market currently contain a relatively low concentration of the drug, as the usual dose of simvastatin in marketed formulations is 20 mg/day (doses range from 10 to 80 mg/day). Authors using spray-dried emulsions containing simvastatin or its analogues reported achieving a drug content of approximately 15 to 29 mg/g. However, their final drug delivery systems were mainly dry emulsions that were further processed into various solid dosage forms [51,52,53]. In our study, the theoretical drug content ranged from 41.7 mg/g to 76.9 mg/g depending on the antiadhesive used. However, the practical drug content was slightly lower and depended on the encapsulation efficiency, which also varied depending on the antiadhesive used and the location of particle sampling (Figure 3). When we compare the particles without the addition of AA, we find that the drug content was highest when CaSi was used as the antiadhesive, with 61.7 mg simvastatin per gramme of dry matter when sampled from the collection vessel and 56.1 mg/g when sampled from the cyclone. In the case of NCC/lactose and lactose alone, the drug content was significantly lower, as the content of antiadhesive in the initial emulsion was twice as high as in the case of CaSi, namely for the lactose 29.1 mg/g (from the collection vessel) and 15.3 mg/g (from the cyclone) while 36.8 mg/g (from the collection vessel) and 15.3 mg/g (from the cyclone) for NCC/lactose as antiadhesive.

### 3.5. Chemical Stability of Simvastatin in Dry Particles

When examining the chemical stability of the drug during a one-month storage, our results showed that simvastatin was best preserved when incorporated into CaSi particles, removed from the collection vessel, and stored under room conditions (Figure 4). This was expected in part because CaSi is known to form a protective barrier around the drug, which not only increases stability during the process and one-month storage, but also ensures the prolonged release of the drug in the later stages [54]. However, the concentration of simvastatin in the lactose and NCC/lactose particles was significantly lower after one month of storage, especially in the collection vessel samples, suggesting oxidative, and/or hydrolytic degradation of simvastatin during storage.

Simvastatin in the dry particles was in an amorphous solid state immediately after spray drying and after one month of storage at both room (25 °C, 40% RH) and accelerated (40 °C, 75% RH) storage conditions (Appendix A), regardless of the antiadhesive used, whereas simvastatin in a powder was in a crystalline form (Appendix A). Previously published results showed relatively high chemical stability of pure simvastatin after 30 days under accelerated conditions compared to simvastatin containing particles [30], indicating a better chemical stability of crystalline drugs compared to their amorphous forms, as already reported by some other authors [55]. In agreement with this, we can observe that our spray-dried drug-containing particles containing amorphous simvastatin showed relatively low chemical stability after one month of storage, which was later increased by the addition of an antioxidant.

By adding AA to the initial formulation, we observed that the stability of simvastatin increased sharply in the case of lactose and NCC/lactose particles, regardless of the sampling location and storage conditions. Thus, in the case of lactose, the concentration of simvastatin in the dry particles stored under room conditions was more than 80% (88.22% in the cyclone and 82.94% in the collection vessel of the initial drug content), whereas under accelerated storage conditions, 28.54% of the drug was retained in the cyclone and 61.31% in the collection vessel, respectively. After the addition of AA, the highest active ingredient content was observed in the NCC/lactose particles, where more than 85% of the active ingredient was retained at both sampling locations under room conditions. However, under the accelerated conditions, only 50% of the drug was retained. The result was also rather reversed for CaSi particles, as the presence of AA visibly decreased the simvastatin content after one month of storage under both storage conditions, whereas significance was only detected for particles from the collection vessel. The explanation for these phenomena is rather difficult. However, it is known and has been reported previously that the CaSi itself can cause oxidative degradation of the materials [56], but then, this should also be observed as low drug stability in CaSi particles without AA addition. However, our results prove just the opposite. Moreover, calcium ascorbate may also be formed during the process. However, the antioxidant activity of calcium ascorbate has been shown to be the same as that of AA, which means that it should not decrease the chemical stability of simvastatin over the course of a month [57].

### 3.6. Moisture Content

Since water content is one of the most important factors inducing the hydrolysis of simvastatin, we monitored the moisture content in all particles with and without AA after one month of storage. The moisture content ranged from 0.67 to 1.78%, depending on the type of antiadhesive used and the presence of AA. The lowest moisture content was found in NCC/lactose particles without AA (0.67%), with no significant difference from particles with AA (0.70%). However, the highest moisture content was measured in CaSi particles with AA (1.78%) while particles without AA contained only 0.81% moisture. This could possibly influence the degradation of simvastatin during storage period, which was highest in CaSi AA particles. 

In addition, the content of simvastatin hydroxyacid in the dry particles was determined in all experiments. Namely, simvastatin is a lactone prodrug in the form of a cyclic ester that hydrolyzes in the presence of water to its biologically active metabolite, simvastatin hydroxyacid [30]. The content of simvastatin hydroxyacid was insignificantly low (<0.5 µg/mL) in most experiments, except in the CaSi particles (all from the collection vessel). Particles without AA, stored at 25 °C, contained 0.92 µg/mL while the same particles with AA contained 4.3 µg/mL of hydroxyacid. Similarly, the hydroxyacid content in dry particles stored at 40 °C with AA was almost twice as high (7.48 µg/mL) as in particles without AA (4.68 µg/mL). 

### 3.7. Redispersion of Dry Particles in Blank Ink Medium

After preparing the dry particles, we dispersed them in three different pre-prepared blank ink media consisting of PG and water mixtures (10/90, 50/50, and 90/10% (*v*/*v*)) with different viscosity and surface tension properties. The dynamic viscosity of the prepared media ranged from 1 to 26 mPa.s while the surface tension was approximately between 35 and 57 mN/m (Figure 5), which is consistent with the suitable properties for successful inkjet printing published in the literature [58,59,60,61]. According to the authors, the viscosity and surface tension of the ink are two of the most important properties for the optimal functioning of inkjet printing [62]. With this in mind, we have selected three different blank ink media to cover the full range of desired properties for successful inkjet printing and to find the most suitable solution for our purpose.

Redispersion of the particles was performed according to two different procedures. In the first procedure, the particles were simply added to the selected medium and mixed manually while in the second procedure, we mixed the system with a vortex for an additional two min. For all samples, the first of the three parallels was tested first to determine the effects of the different mixing procedures on the final particle size. If no size difference was found, vortex mixing was excluded for the next two parallels. For each of these three parallels, the particle size (d10, d50, and d90) was measured.

#### 3.7.1. Redispersion of Particles with Lactose as Antiadhesive

The particle size measured after redispersion in Figure 6 shows a close resemblance to the particle size on the images from SEM (Figure 2a). This indicates the presence of particle clusters in the medium and that the clusters were neither broken nor enlarged during redispersion. The size of the dry particles (Table 3), on the other hand, was much smaller compared to the freshly redispersed particles, confirming our theory about the breakup of the clusters due to the high pressure during the measurements of the dry particles.

However, we can see some differences in particle size when we use different time frames, mixing methods, and different redispersion media.

Figure 6 shows that the particle size of all samples with lactose as an antiadhesive decreased significantly during the one-month storage. This indicates that the additional agglomeration, and thus the size of the measured particles, does not occur in the medium, as it would only increase with time. We attribute this result mainly to the dissolution of lactose in the medium during the one-month storage, which surprisingly was highest in the most viscous medium, where the smallest particles of all were obtained. It is also observed that vortex mixing has a significant effect on the reduction of particle size after redispersion for all samples, regardless of the sampling location. For example, 2 min of additional vortex mixing results in particles smaller than 10 µm for all measurements after 1 month of storage. Most likely, partial mechanical breakup of the clusters or simply accelerated dissolution of the lactose occurred during vortex mixing and continued over the course of a month.

After one month, the size of the vortex-mixed particles from the collection vessel was 6.93 µm, which is close to the size of the dry particles (5.36 µm). A surprising result is the size of the particles from the cyclone that were additionally mixed with a vortex after redispersion in the most viscous medium (90 vol% PG). The measured size of these particles after one month was 5.67 µm, which is even lower than the size measurement of the same dry particles (6.11 µm). Besides the additional lactose dissolution, the reason could be that the larger particles in the cyclone essentially remain on the walls of the cyclone and move slowly toward the collection vessel while the smaller particles (about 1 μm) are distributed throughout the volume of the cyclone, depending on the airflow itself. Therefore, in the case of collisions with large particles, they can also stick and deposit on the cyclone walls. This results in larger dry particles on the walls of the cyclone on the one hand and smaller particles on the other, the size of which is measured after their redispersion. The reason for this is that the bonds between these particles are most likely very weak, so the particles break down into smaller particles after their redispersion in the medium [63].

#### 3.7.2. Redispersion of Particles with NCC/Lactose as Antiadhesive 

When dispersing the NCC/lactose particles collected from the cyclone, we did not use vortex mixing because the initial experiments showed no significant difference in particle size. 

First, comparing the size of the dry particles measured by laser diffraction (5–7 μm) with the size of redispersed particles, a very large size difference occurs. After redispersion, nanometer range particles were obtained (330–690 nm), which is about 15 times smaller than the same dry particles (Figure 7). The explanation most likely relates firstly to the breakup of the clusters in the medium and secondly to the separation of the NCC from the drug and polymer containing particles, which significantly influenced the additional size reduction (compared to the size of the dry particles, which were also subjected to the cluster breakup).

Comparison of samples from the collection vessel with and without vortex mixing showed that particle size was slightly reduced in most cases when the vortex was used (except in the case of the 50/50 composition). The most significant difference in particle size was in the most viscous medium, where the smallest particles were obtained (350 nm). Here, we note that the effect of the viscosity of the medium on particle size is similar to that observed for lactose. Furthermore, when we compare the size of the dispersed particles before and after one month of storage, we find that the size of the dispersed particles changed significantly only in the case of the 90/10 composition, where the particle size is smaller, and the 50/50 composition of the medium, where the particle size increased dramatically. This could be related to the previous observation, which is related to the separation of the antiadhesive from the polymer particles, leading to their agglomeration after one month of storage. Possible agglomeration can be explained by comparing d90 of the particles (Figure 8). It was found that in most cases the polymer particles start to agglomerate in the course of one month, which is most evident for the particles from the collector in a medium containing 50% (*v*/*v*) water (processing the sample with a vortex further increases particle agglomeration).

#### 3.7.3. Redispersion of Particles with CaSi as Antiadhesive

When dispersing the CaSi particles from the collection vessel, we did not use vortex mixing because the initial experiments showed no significant difference in particle size. 

When we compare the size of the dry particles (14–21 µm) with CaSi as an antiadhesive, the size of the dispersed particles from the collection vessel increased 2.8 times and that of the particles from the cyclone increased 1.3 times compared to the dry particles (Figure 9). This result is most likely due to the agglomeration of the particles in the medium.

Mixing with a vortex contributed to a decrease in particle size in all cases (except for the samples from the collection flask). The largest difference in the direction of particle reduction is observed in the least viscous medium, where the size of the particles decreases by more than 5 μm when the vortex is used, while in the medium with 50 vol% PG the particles decrease by only 2 µm. It can be seen that as the viscosity of the medium increases, so does the size of the particles dispersed with the vortex. This is surprising from the point of view of CaSi as an antiadhesive, since one would expect the water content in the medium to affect the poorer dissolution of the antiadhesive and, consequently, the size of the particles themselves in the medium.

In conclusion, the results of our study show that among the other tested antiadhesives, the addition of NCC/lactose to the initial emulsion formulation is the most suitable in terms of the size of the dispersed particles. Thus, the prepared particles could be suitable for inkjet printing technology, as the physical stability of the particles indicates that NCC/lactose particles can be used in printing ink without causing undesirable nozzle clogging (depending on the size of the print head). To confirm the above statement, inks prepared in this way with NCC/lactose particles as ink medium should be tested with a suitable inkjet printer.

## 4. Conclusions

In the present study, we have shown that spray drying of organic emulsions can serve as a method to produce drug/polymer particles suitable for ex tempore preparation of 2D printing cartridges after their redispersion in a suitable blank ink medium. Using DoE, we have developed a model that proposes parameters for the spray drying process that led us to the smallest particles, whose morphology differs depending on the antiadhesive used. The prepared drug-loaded particles containing different antiadhesives with and without antioxidants AA were evaluated for the stability of simvastatin in the dry formulation at two different sampling locations immediately after the process and after one month of storage. It was found that the addition of antioxidants to the particle formulation processed by spray drying significantly affected the oxidative stability of the drug, but surprisingly very differently for the different antiadhesives. Namely, the stability of the drug increased for lactose and NCC/lactose particles, while it decreased significantly for CaSi particles, indicating that antiadhesives should be carefully selected for the intended purpose. Dispersion of three different types of dry particles in three different PG and water mixtures, which formed a printable blank ink medium, revealed that NCC/lactose was the most suitable antiadhesive in terms of physical stability of such particles in different blank ink media. Even after one month of storage, the redispersed NCC/lactose particles remain in the same size range without any undesirable particle aggregation. After one month of storage, the NCC/lactose particles prepared in this manner represent a dry suspension with satisfactory chemical stability of the drug and exhibit adequate size and physical stability after their subsequent redispersion in a suitable printable blank ink medium. With following long-term stability testing, the NCC/lactose dry particles could possibly have the potential to be used in 2D printing technology as a pre-formulation for the production of printable inks ex tempore.

## Figures and Tables

**Figure 1 pharmaceutics-15-02221-f001:**
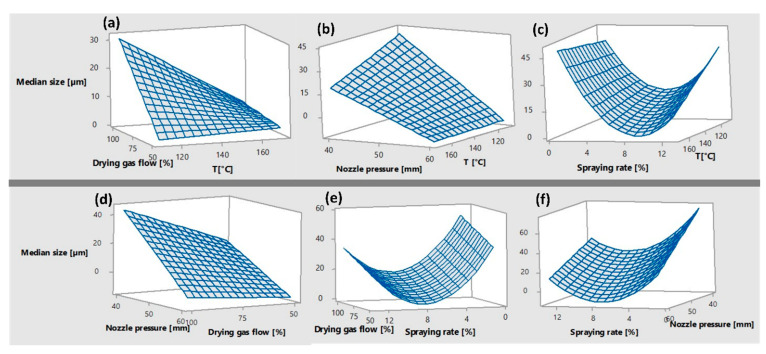
Surface plot showing the effects of (**a**) drying gas flow and temperature, (**b**) nozzle pressure and temperature, (**c**) spraying rate and temperature, (**d**) nozzle pressure and drying gas flow, (**e**) drying gas flow and spraying rate, and (**f**) spraying rate and nozzle pressure on median particle size.

**Figure 2 pharmaceutics-15-02221-f002:**
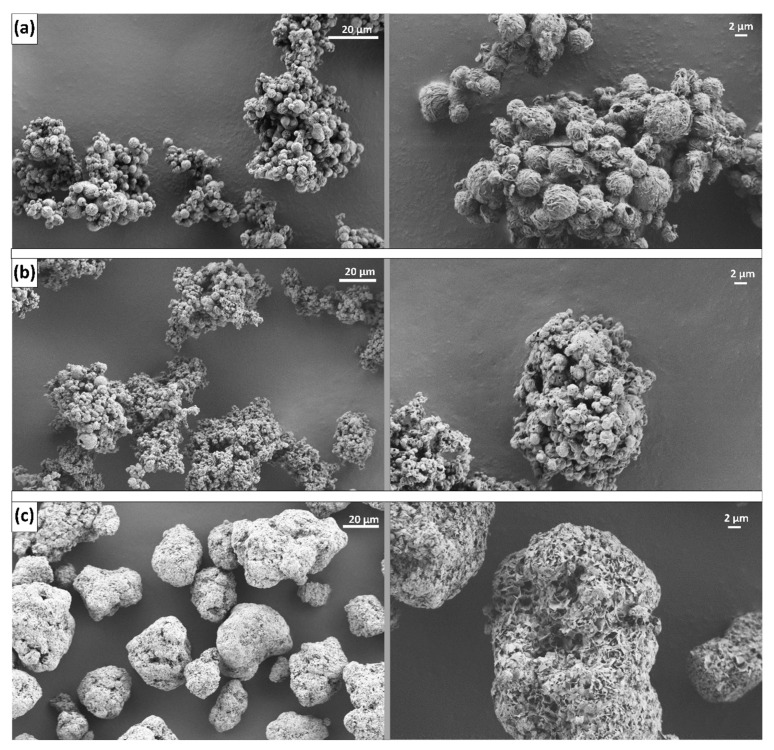
Morphology of particles containing (**a**) lactose, (**b**) NCC/lactose = 25:75% (*w*/*w*), and (**c**) CaSi as antiadhesive substances.

**Figure 3 pharmaceutics-15-02221-f003:**
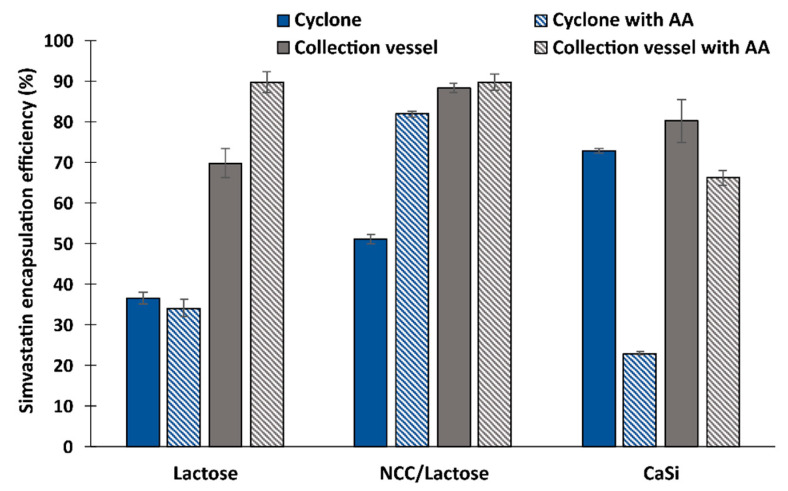
The encapsulation efficiency of simvastatin, determined by UPLC analysis, of three different types of particles taken from two different locations (cyclone, collection vessel) with and without AA addition.

**Figure 4 pharmaceutics-15-02221-f004:**
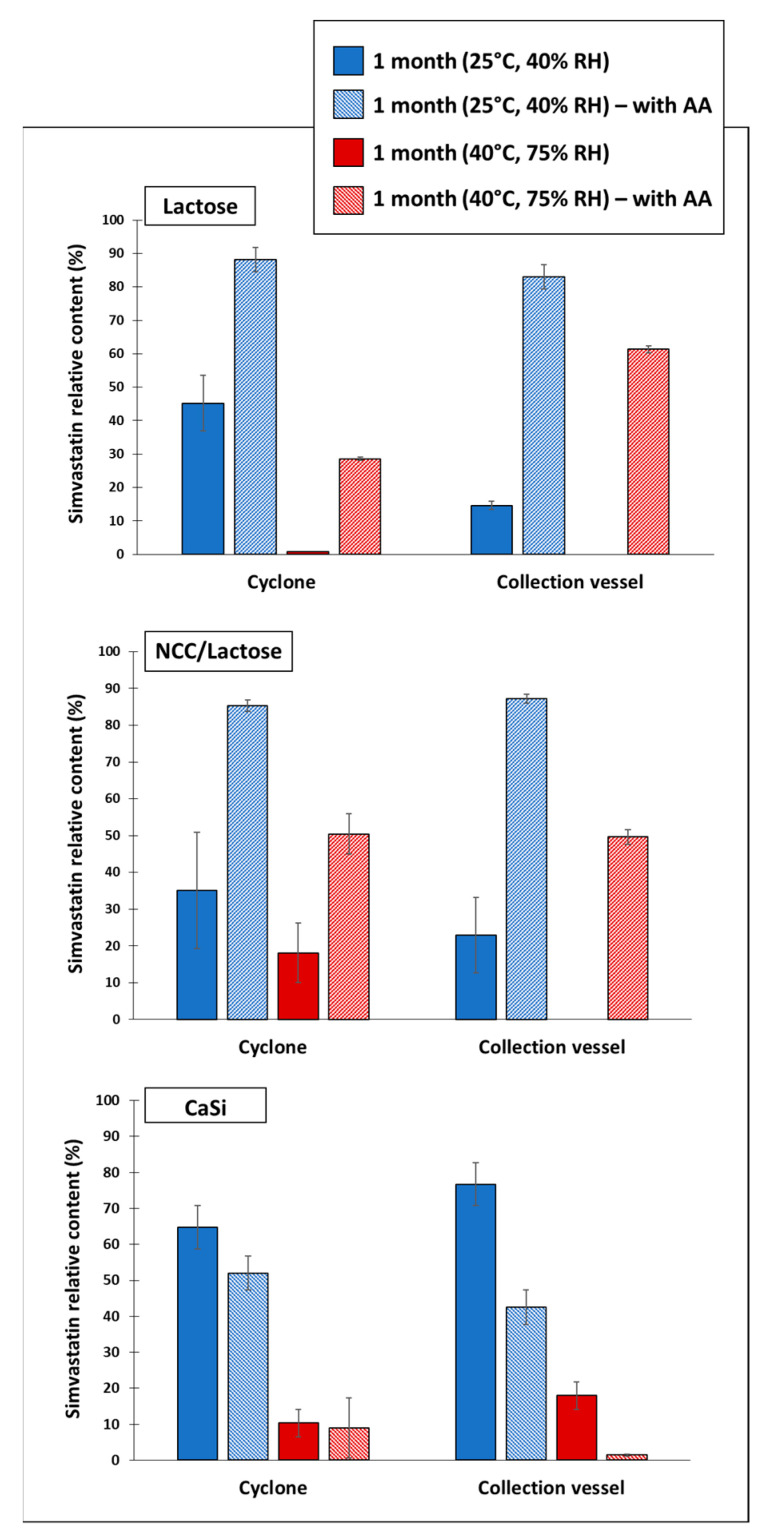
Simvastatin content relative to after production content after one month at room (25 °C, 45 RH) and accelerated conditions (40 °C, 75% RH), determined for three different types of particles with and without ascorbic acid (AA), sampled from cyclone and collection vessel.

**Figure 5 pharmaceutics-15-02221-f005:**
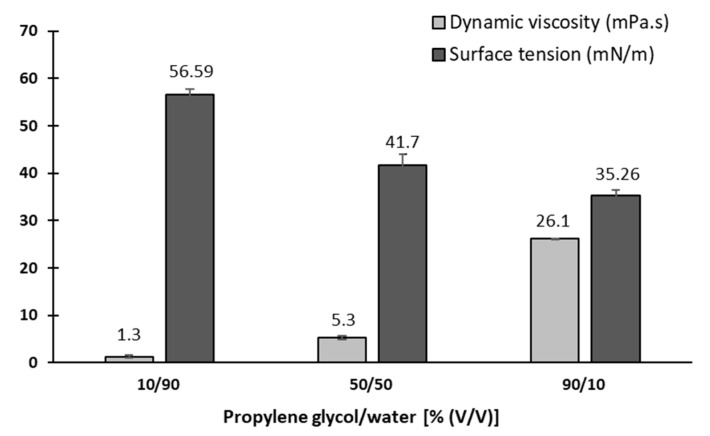
Surface tension and dynamic viscosity in three different blank ink media solutions.

**Figure 6 pharmaceutics-15-02221-f006:**
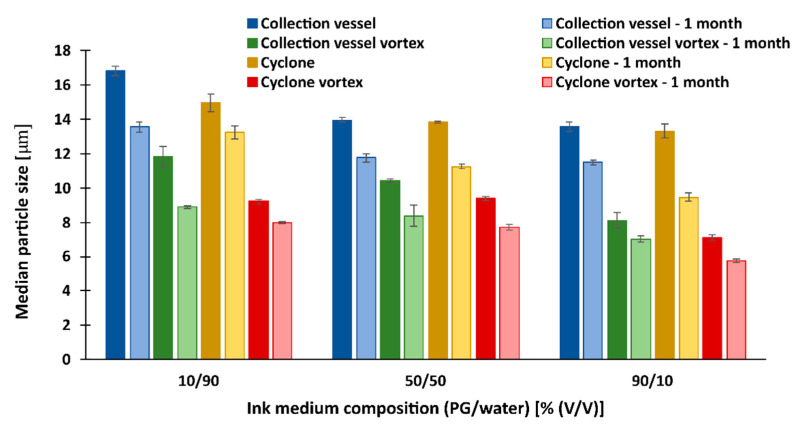
The median size (d50) of redispersed particles in different media, with lactose as antiadhesive substance.

**Figure 7 pharmaceutics-15-02221-f007:**
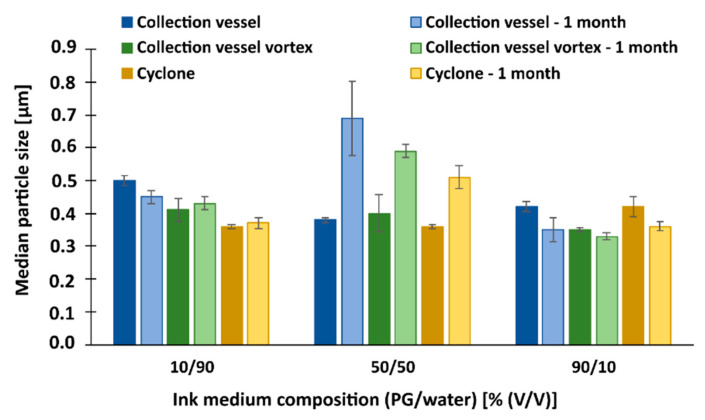
The median size (d50) of redispersed particles in different media, with NCC/lactose as antiadhesive substance.

**Figure 8 pharmaceutics-15-02221-f008:**
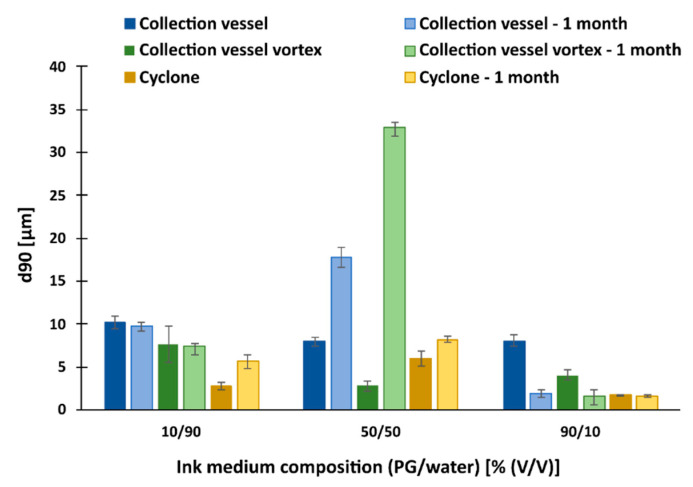
d90 of particle size, redispersed in three different media, with NCC/lactose as antiadhesive substance.

**Figure 9 pharmaceutics-15-02221-f009:**
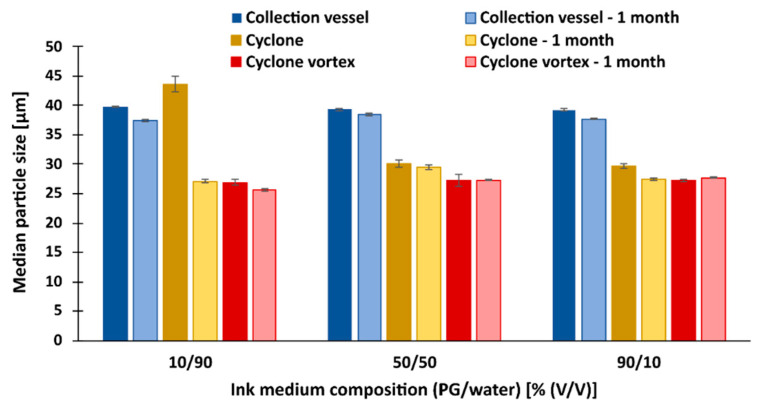
The median size (d50) of redispersed particles in different media, with CaSi as antiadhesive substance.

**Table 1 pharmaceutics-15-02221-t001:** Emulsion droplet sizes and PDI at different water to chloroform ratios.

Emulsion	H_2_O to CHCl_3_ Ratio (*w*/*w*)	Droplet Size [nm]	PDI
EMU 1	4:1	216.2 ± 11.3	0.118 ± 0.024
EMU 2	3:1	253.9 ± 9.4	0.179 ± 0.017
EMU 3	5:2	289.0 ± 9.7	0.258 ± 0.021
EMU 4	2:1	342.3 ± 10.2	0.268 ± 0.018

**Table 2 pharmaceutics-15-02221-t002:** Surface response design with five independent variables: inlet temperature, drying gas flow (% of aspiration rate), nozzle pressure (flow meter spraying air), spraying rate (% of maximum peristaltic pump rotation), and nozzle geometry setting *, and three responses: d(50) particles in collector, d(50) particles in cyclone, and d(50) particles of combined product. The dark gray areas indicate the smallest particles produced.

Exp	Inlet Temperature [°C]	Drying Gas Flow [%]	Nozzle Pressure [mm]	Spraying Rate [%]	Nozzle Geometry [mm]	d(50) Collector [μm]	d(50) Cyclone [μm]	d(50) Combined Product [μm]
E1	140	75	50	7	0.375	12.99	10.31	11.08
E2	170	100	60	1	0.75	22.25	33.65	33.65
E3	140	100	50	7	0.375	14.72	10.31	11.03
E4	140	75	60	7	0.375	11.85	8.61	8.69
E5	110	50	40	1	0	98.77	74.06	74.06
E6	110	50	40	1	0.75	97.01	53.28	62.28
E7	110	100	40	1	0.75	124.22	83.24	80.11
E8	110	50	40	13	0	347.39	25.63	25.63
E9	140	75	50	1	0.375	30.58	38.73	32.26
E10	170	100	40	13	0	19.7	11.21	12.10
E11	110	100	40	1	0	65.4	70.27	70.27
E12	140	75	50	7	0.75	15.66	11.16	11.61
E13	110	100	60	1	0.75	36.06	20.51	18.37
E14	170	75	50	7	0.375	7.76	7.35	7.93
E15	170	100	40	1	0	52.66	92.53	92.53
E16	110	75	50	7	0.375	6.93	12.48	12.48
E17	170	100	40	1	0.75	44.29	80.43	80.43
E18	170	50	60	1	0	17.03	15.41	16.98
E19	110	100	40	13	0.75	50.79	68.07	68.07
E20	140	50	50	7	0.375	107.69	10.82	11.46
E21	170	50	60	1	0.75	16.84	13.83	13.73
E22	110	50	60	1	0	21.88	13.94	16.15
E23	170	50	40	13	0.75	71.64	15.19	25.48
E24	170	100	60	13	0.75	22.26	7.96	8.68
E25	110	100	60	1	0	37.27	23.87	20.73
E26	170	50	40	1	0	29.08	54.27	54.27
E27	140	75	50	7	0	7.28	8.01	8.02
E28	170	50	40	1	0.75	62.84	83.86	83.86
E29	170	100	40	13	0.75	11.97	9.44	10.49
E30	140	75	40	7	0.375	12.29	11.05	11.33
E31	170	50	40	13	0	29.19	11.25	17.49
E32	110	50	60	13	0.75	300.69	12.08	12.08
E33	110	50	40	13	0.75	155.93	12.45	12.45
E34	170	50	60	13	0	86.57	12.24	12.71
E35	140	75	50	7	0.375	13.97	10.88	11.35
E36	110	50	60	1	0.75	17.51	12.57	12.95
E37	110	50	60	13	0	295.33	8.12	8.12
E38	140	75	50	13	0.375	8.8	9.77	9.36
E39	140	75	50	7	0.375	9.58	8.71	9.07
E40	110	100	60	13	0	60.2	24.93	24.93
E41	140	75	50	7	0.375	8.51	10.38	9.29
E42	170	100	60	1	0	9.94	27.83	20.76
E43	110	100	40	13	0	14.11	163.98	163.98
E44	170	100	60	13	0	9.4	8.05	7.61
E45	110	100	60	13	0.75	72.31	20.03	20.03
E46	170	50	60	13	0.75	65.44	8.1	10.55
E47	170	84.85	40	13	0.3788	10.37	8.98	9.15
E48	170	100	60	8.39	0.2273	5.89	6.60	6.70

* The nozzle geometry setting is defined as 0 mm when the nozzle tip is parallel to the nozzle cap and 0.75 mm when the nozzle cap is fully screwed in, as previously reported by Pohlen et al. [29].

**Table 3 pharmaceutics-15-02221-t003:** The size of dry particles containing different antiadhesives taken from two different locations.

Antiadhesives	Sampling Location	d10 [μm]	d50 [μm]	d90 [μm]	SPAN
Lactose	Collection vessel	1.17 ± 0.28	5.36 ± 0.71	20.56 ± 2.41	3.62 ± 0.11
	Cyclone	1.30 ± 0.45	6.11 ± 0.96	34.71 ± 2.78	5.47 ± 0.23
NCC/Lactose	Collection vessel	1.15 ± 0.37	5.47 ± 0.88	15.76 ± 1.71	2.67 ± 0.08
	Cyclone	1.38 ± 0.38	7.22 ± 1.01	18.71 ± 3.02	2.40 ± 0.24
CaSi	Collection vessel	1.96 ± 0.28	14.81 ± 0.11	44.56 ± 0.56	2.88 ± 0.15
	Cyclone	2.00 ± 0.24	21.51 ± 0.65	64.71 ± 0.78	2.92 ± 0.12

## Data Availability

The data presented in this study are available upon request from the corresponding author.

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
