# Peer review of "Development of Simvastatin-Loaded Particles Using Spray Drying Method for Ex Tempore Preparation of Cartridges for 2D Printing Technology"

_pharmaceutics, 2023, doi:10.3390/pharmaceutics15092221_

Round 1

Reviewer 1 Report

The manuscript titled “Development of simvastatin-loaded particles using spray drying method for ex tempore preparation of cartridges for 2D printing technology” describes the preparation of inkjet inks made out of spray dried particles.

I believe the manuscript lacks direction and details. The introduction requires particular revision as the first half seems to have been rephrased from previous manuscripts and does not fit the current work which focuses on inkjet 2D printing which is not even defined or explained. Nor do the authors talk about current inks and pharmaceutical inks. Also, the manuscript lacks details and in most cases shows singular data rather than data averaged from repetitions, which is not scientifically sound.

More detailed comments:

-       The abbreviation PG is used in the intro without being introduced.

-       v/v should not be capitalised.

-       Line 50: space missing before reference [11].

-       “The compact size of inkjet printing equipment enables manufacturing of medicine upon request at the dispensing site, such as a pharmacy or hospital, with at-line non-destructive spectroscopic techniques (NIR) used to verify the drug assay on site.” Firstly, inkjet printers are not compact and some are incredibly large. Moreover, NIR is an abbreviation and the words that precede it are not its definition.

-       “However, it should be noted that nano/micro-particle dispersions are known to be physically unstable systems, which can be an important challenge to overcome from the point of view of printability and non-uniform drug delivery. However, these challenges can be faced by using techniques providing nearly monodispersed particles in size and shape and with that diminish the possibility of instability of such systems.” Although they are in different paragraphs, it is grammatically incorrect to have 2 sentences starting with “however” follow each other. Please revise.

-       The authors use abbreviations without actually introducing them first. E.g., SIM, PCL, IKA...etc.

-       SEM: what about coating? Did you coat the particles with anything (e.g., gold) before imaging? If so, it must be mentioned in details.

-       Table 1: EMU 3; I can see 3 ratios but it is unclear what is the 3rd ratio? Also, the measurements should be done at least in triplicates, displayed in average and include a standard deviation for droplet size and PDI.

-       It is unclear what the content of EMU 1, 2, 3 and 4 is. This must be clarified using another table.

-       Figure 1: the figure legend should include the a, b, c, d, e and f labels.

-       Table 3: again, this should be an average of at least 3 repetitions and you should include the standard deviations.

-       “Moreover, simvastatin was in the amorphous solid state in the dry particles right after spray drying and one-month of storage under both storage conditions, regardless of the antiadhesive used (data not shown).” If the authors don’t want to show the data in the manuscript, this should be shown as supplementary data.

-       Figure 5: average + standard deviation?

-       Some of the text has a different font size (sometimes bigger and sometimes smaller – figures’ legends particularly).

-       The heading references is misspelled.

-       The references are old and outdated. Authors must update their reference list with more recent and relevant references.

Revise English language.

Some comments shown above.

Author Response

CORRECTIONS MADE TO MANUSCRIPT ID: pharmaceutics- 2562134

Title: Development of simvastatin-loaded particles using spray drying method for ex tempore preparation of cartridges for 2D printing technology

We would like to thank the Reviewer for the thoughtful comments and detailed evaluation of our manuscript. We have thus revised and improved the manuscript in accordance with these comments. More specifically, the following changes (see our point-by-point replies below) and corrections to the original manuscript have been made, and we hope that the revised manuscript will answer the comments sufficiently.

ANSWERS to the Reviewer 1

The manuscript titled “Development of simvastatin-loaded particles using spray drying method for ex tempore preparation of cartridges for 2D printing technology” describes the preparation of inkjet inks made out of spray-dried particles.

I believe the manuscript lacks direction and details. The introduction requires particular revision as the first half seems to have been rephrased from previous manuscripts and does not fit the current work which focuses on inkjet 2D printing which is not even defined or explained. Nor do the authors talk about current inks and pharmaceutical inks. Also, the manuscript lacks details and in most cases shows singular data rather than data averaged from repetitions, which is not scientifically sound.

Response: We thank the Reviewer for the comments. We have supplemented and rewritten the introduction of the manuscript and included more recent and relevant literature (see the revised manuscript).

Regarding the presentation of data, all experiments were performed in triplicate, but we agree that they were not adequately presented. We have replaced/added the corresponding figures and tables by adding the standard deviation values that were not included in the submitted version. We have also included the triplicate information in the method section where relevant. We sincerely apologize for the error and hope that this correction sufficiently improves the presentation of the data.

Point 1: The abbreviation PG is used in the intro without being introduced.

Response: We thank the Reviewer for the observation. We have added the abbreviation explanation accordingly.

Point 2: v/v should not be capitalized.

Response: We have corrected it accordingly throughout the manuscript.

Point 3: Line 50: space missing before reference [11].

Response: The space has been inserted in the text.

Point 4:The compact size of inkjet printing equipment enables manufacturing of medicine upon request at the dispensing site, such as a pharmacy or hospital, with at-line non-destructive spectroscopic techniques (NIR) used to verify the drug assay on site.” Firstly, inkjet printers are not compact and some are incredibly large. Moreover, NIR is an abbreviation and the words that precede it are not its definition.

Response: We thank the Reviewer for the comment. The statement has been corrected in the text and the abbreviation explained.

Point 5: “However, it should be noted that nano/micro-particle dispersions are known to be physically unstable systems, which can be an important challenge to overcome from the point of view of printability and non-uniform drug delivery. However, these challenges can be faced by using techniques providing nearly monodispersed particles in size and shape and with that diminish the possibility of instability of such systems.” Although they are in different paragraphs, it is grammatically incorrect to have 2 sentences starting with “however” follow each other. Please revise.

Response: With the rewriting of the introduction, we have also grammatically corrected the problem highlighted in the Reviewer's comment.

Point 6: The authors use abbreviations without actually introducing them first. E.g., SIM, PCL, IKA...etc.

Response: We thank the Reviewer for pointing this out. We have added all abbreviation explanations accordingly.

Point 7: SEM: what about coating? Did you coat the particles with anything (e.g., gold) before imaging? If so, it must be mentioned in details.

Response: For SEM imaging, we used a very low acceleration voltage (1.0 kV) for which coating is not necessary. Therefore, we did not coat the samples prior to imaging. Our sample preparation consisted only of placing the particles on a carbon tape and examining them with SEM as described in Section 2.2.4.

Point 8: Table 1: EMU 3; I can see 3 ratios but it is unclear what is the 3rd ratio? Also, the measurements should be done at least in triplicates, displayed in average, and include a standard deviation for droplet size and PDI.

Response: The 3rd ratio is 2,5:1. For more clarity, we have changed it to a ratio of 5:2. The measurements were done in triplicates, but as mentioned above, they were not presented correctly in the table (standard deviations are missing). We have corrected the emphasized error in the revised manuscript.

Point 9: It is unclear what the content of EMU 1, 2, 3 and 4 is. This must be clarified using another table.

Response: All four final emulsions consisted of an organic (0.8 g SIM, 1.6 g PCL in chloroform) and a water phase (4 g Tween® 20 in water) as described in the Method section. We sincerely appreciate the Reviewer's suggestion regarding the inclusion of an additional table. Nevertheless, we hold the belief that the revised text should provide readers with a clear understanding since all four emulsions contain the same amount of the same solid components. We sincerely hope that the Reviewer finds our rationale for this decision acceptable.

For more clarity, we revised the text in the method section (2.2.1.) by adding some additional information:  “The four emulsions listed in Table 1 (EMU 1, EMU 2, EMU 3, and EMU 4) consist of an organic phase (0.8 g SIM, 1.6 g PCL in chloroform) and a water phase (4 g Tween® 20 in water), with the most optimal emulsion being EMU 2, which contains 40 g chloroform and 120 g water.”

We also replaced the sentence: “Moreover, the amount and type of surfactant as well as the ratio between the inner and outer phases of the emulsion were adjusted during the preliminary optimization of the emulsion,” making only confusion to the readers in section 3.1. (Results and discussion), with the sentence: “After preliminary optimization of the emulsion, the emulsion described in section 2.2.1. was examined for different ratios between the inner and outer phase (Table 1).”

We very much hope that with these corrections we have given the readers more clarity about the composition of the emulsion and that the Reviewer finds this acceptable.

Point 10: Figure 1: the figure legend should include the a, b, c, d, e, and f labels.

Response: The requested captions were added to the description of the figure.

Surface plot showing the effects of (a) drying gas flow and temperature, (b) nozzle pressure and temperature, (c) spraying rate and temperature, (d) nozzle pressure and drying gas flow, (e) drying gas flow and spraying rate, (f) spraying rate and nozzle pressure on median particle size.”

Point 11: Table 3: again, this should be an average of at least 3 repetitions and you should include the standard deviations.

Response: We again sincerely apologize for the error and thank the Reviewer for noticing it. The measurements were again performed in triplicate without being listed in the table. We have added the missing standard deviations in the revised manuscript.

Point 12: Moreover, simvastatin was in the amorphous solid state in the dry particles right after spray drying and one-month of storage under both storage conditions, regardless of the antiadhesive used (data not shown).” If the authors don’t want to show the data in the manuscript, this should be shown as supplementary data.

Response: The DSC diagrams along with the DSC method and short description of the results were included in the supplementary data, as requested. We decided not to include them in the manuscript only because of its length. We very much hope that the Reviewer finds this acceptable.

Point 13: Figure 5: average + standard deviation?

Response: The missing standard deviations were added to the chart.

Point 14: Some of the text has a different font size (sometimes bigger and sometimes smaller – figures’ legends particularly).

Response: The font size was corrected according to the requirements of the Journal.

Point 15: The heading references is misspelled.

Response: We thank the Reviewer for this remark. It has been corrected accordingly.

Point 16: The references are old and outdated. Authors must update their reference list with more recent and relevant references.

Response: We have included more recent and relevant literature, especially in the introduction to the manuscript, as mentioned above.

Point 16: Revise English language.

Response: We have revised the English throughout the manuscript, but have not highlighted all the changes made, as this would detract from the clarity of the substantive changes made at the request of the Reviewers. We very much hope that the reviewer finds this acceptable.

Reviewer 2 Report

The manuscript describes an interesting study performed by Sterle Zorec and Dreu. Please find my comments below:

General terms:

-          There are quite a few typos in this text. Please check for spelling, a few are mentioned below.

Comments:

-          Line 19: PG was not introduced before in this study

-          Line 73, 74: “we propose for the first time a solution to the problem of physical and chemical instability of the final ink formulations”. The approach is very interesting, but samples were not stored long enough and only at room conditions and not at conditions according to the European Pharmacopeia. These investigations do not allow this kind of statement for the ink formulations.

-          Line 94: propilene glycol (PG), please correct typo

-          Line 105: SIM and PCL were not introduced as abbreviations before

-          Line 124: The reference number of the reference Pohlen et. al. is missing

-          Lines 127 ff: the total amount of added antiadhesive substances should be replaced by a concentration (in percent) in the formulation to display a comprehensible formulation composition  

-          Line 225 ff: “However, since the size of the dry particles is directly related to the size of the dispersed particles, we optimized our spray drying process to obtain the smallest dry particles possible”. In general, it was shown later that the particle sizes mainly depend on the antiadhesives added to the formulation. Is it really clear that they directly rely on each other? I do not think that this general statement can be made here. Please comment on this and rephrase the sentence.

-           Line 267 ff: it is said that samples were collected from different parts (vessel and cyclone) and combined and sizes were measured. Please state which amounts of powder were collected from the vessel and the cyclone.

-          Line 267 ff: in terms of scale-up: how should a powder collection of this formulation should work in an industrial scale? Please comment on this.

-          Chapter 1.1 The size of the dry particles after spray-drying: DoE approaches are interesting but, in this case, it becomes clear that they cannot describe such a complex system. 48 experiments were proposed by the DoE-program and performed but the discussions starting from Line 303 showed that this DoE does not result in a clear statement, e.g. in regards of temperature influence. In the end, the formulation resulting in the smallest particle sizes E48 was chosen but without clearly understanding the influences and further optimizing the parameters based on the results of the DoE. This chapter has to be revised and restructured and it has to be considered if this DoE approach could really serve as a basis for this parameter optimization.

-          Figure 3: please add lines for the numbers on the y-axis

-          Figure 3: clear differences in the encapsulation efficiency (without AA and with AA) were shown in regards to the location of the sampling. What is the reason for this? Please discuss.

-          Line 414: typo “lover”

-          Line 438: “both storage conditions”: what storage conditions were used regardless of room temperature? This information is only given in graph 4. Please add this information in the methods-section and in the text.

-          Line 434: “Larger particles most likely indicate a higher amount of antiadhesive in the particles and thus around the drug, which may protect it from degradation, as shown by our results”. Why is there a higher amount of antiadhesive in larger particles? The ratio of the drug to antiadhesive should be the same in the whole formulation.

-          Line 439: it is not clear why the amorphous solid state of the particles has t result in a lower chemical stability of the drug after only one month. Please comment on this.

-          Figure 4: why was the drug content so low for samples with lactose, storage for 1 month at 40 °C?

-          Line 506: please rephrase this sentence for better clarification.

-          Lines 609 ff: this DoE does not allow a detailed discussion of optimized parameters. See comment above.

-          Line 626: The storage times were not long enough to enable a statement regarding chemical and physical stability. In this case it has to be mentioned in the conclusion that the study was only performed over 1 month.

There are quite a few typos in this text. Please check for spelling, a few are mentioned in the comments.

Author Response

CORRECTIONS MADE TO MANUSCRIPT ID: pharmaceutics- 2562134

Title: Development of simvastatin-loaded particles using spray drying method for ex tempore preparation of cartridges for 2D printing technology

We would like to thank the Reviewer for the thoughtful comments and detailed evaluation of our manuscript. We have thus revised and improved the manuscript according to these comments. More specifically, the following changes (see our point-by-point replies below) and corrections to the original manuscript have been made, and we hope that the revised manuscript will answer the comments sufficiently.

ANSWERS to the Reviewer 2

The manuscript describes an interesting study performed by Sterle Zorec and Dreu. Please find my comments below:

General terms:

There are quite a few typos in this text. Please check for spelling, a few are mentioned below.

Response: We thank the Reviewer for the comment. We have revised the spelling throughout the manuscript.

Point 1: Line 19: PG was not introduced before in this study.

Response: We thank the Reviewer for the observation. We have written the full name and explained the abbreviation later in the text.

Point 2:  Line 73, 74: “we propose for the first time a solution to the problem of physical and chemical instability of the final ink formulations”. The approach is very interesting, but samples were not stored long enough and only at room conditions and not at conditions according to the European Pharmacopeia. These investigations do not allow this kind of statement for the ink formulations.

Response: Our goal was to prepare dry particles of a certain physical and chemical stability, which would be added to the blank ink medium right before printing process. Namely, once we add the dry particles to the ink, such final ink formulations will need to exhibit in-use stability only for a week or two (max. 30 days), until the ink is spent. The expected in-use stability period of such inks will depend on (i) the concentration of the drug contained in ink formulation, (ii) on the patient's dose needs and (iii) on the number and the frequency of patients having the same therapy. Thus, our goal is to design dry nano/microparticles based systems as a stable standalone system which will represent a pre-formulation for later ink preparation in a form of suspension, which we explained with the following sentences from the one that the Reviewer has pointed out: “The proposed approach is to prepare the ink formulation ex tempore immediately before its use by simply adding the pre-prepared dry drug-containing particles to the ink base medium, resulting in a particle dispersion suitable for 2D inkjet printing. This "extemporaneous compounding" approach shortens the required shelf life of the classical drug-containing ink formulation, but at the same time extends the shelf life of the dry particle pre-formulation.” And with this statement, we wanted to emphasise that our main focus is on the stability of the dry particles, while the final ink formulations only require some stability until the ink is spent. Our results show that a one-month physical stability of the redispersed NCC/lactose particles (30 days at 25 °C, 45% RH, as we assume that these would be the ambient conditions of a e.g., 2D printer in the hospital), could potentially represent a stable ink system for 2D printing until the ink is spent (30 days at room conditions without agglomeration of the particles). However, we are aware that even to demonstrate the 30-day stability of the ink formulation, we must meet European Pharmacopoeia requirements, which could possibly be confirmed by further long-term stability studies.

As for the stability of the dry particles, in our study we investigated the chemical stability of the drug in the prepared dry particles under room conditions (25 °C and 40% RH) and under accelerated conditions (40 °C and 75% RH), yet only for 1 month. We are aware that the storage period is far below the requirements of the European Pharmacopeia (6 months), which forces us to perform some additional long-term stability studies that we plan to perform in the future.

We have rewritten this particular statement in the introduction and added a few more in the conclusion to provide more clarity. We have also added the description of the stability tests in the Methods chapter, as it was missing in the submitted version of the manuscript. We very much hope that the Reviewer will find this acceptable.

Point 3:  Line 94: propilene glycol (PG), please correct typo

Response: We have corrected the typo. We thank the Reviewer for pointing it out.

Point 4: Line 105: SIM and PCL were not introduced as abbreviations before

Response: We have explained both abbreviations in the Material section where they first appear.

Point 5: Line 124: The reference number of the reference Pohlen et. al. is missing

Response: We have added the reference number in the text.

Point 6: Lines 127 ff: the total amount of added antiadhesive substances should be replaced by a concentration (in percent) in the formulation to display a comprehensible formulation composition.

Response: We thank the Reviewer for the suggestion. We replaced the mass of antiadhesives by their concentration (% (w/v)) in the final emulsion. Namely, the concentration of lactose was 8% (w/v), the combination of NCC and lactose (25:75% (w/w)) was 8% (w/v), while the CaSi concentration was 2.5% (w/v), as indicated in the Method section.

Point 7: Line 225 ff: “However, since the size of the dry particles is directly related to the size of the dispersed particles, we optimized our spray drying process to obtain the smallest dry particles possible”. In general, it was shown later that the particle sizes mainly depend on the antiadhesives added to the formulation. Is it really clear that they directly rely on each other? I do not think that this general statement can be made here. Please comment on this and rephrase the sentence.

Response: By the statement written above, we wanted to express that if the size of the dry particles is small, we also expect the same particles in the dispersion to be small after we placed them into the blank ink medium (unless agglomeration occurs). However, after obtaining the results, it is also clear, as pointed out by the Reviewer, that the same process parameters with different antiadhesives used, lead to different sizes of the dry particles and consequently to different size after their redispersion in the media. With this in mind, we have reworded the statement from: “However, since the size of the dry particles is directly related to the size of the dispersed particles, we optimized our spray drying process to obtain the smallest dry particles possible,” to: “Since we expect the redispersed particles to be in a comparable size range to the same dry particles before redispersion, we have optimized our spray drying process to obtain the smallest possible dry particles,” hoping very much that the Reviewer will find this acceptable.

Point 8: Line 267 ff: it is said that samples were collected from different parts (vessel and cyclone) and combined, and sizes were measured. Please state which amounts of powder were collected from the vessel and the cyclone.

Response: We are aware that process yield is one of the most important factors that should be considered in any manufacturing process, as it determines the efficiency of the process.

In our case, however, we did not include it in our study because our main objective was elsewhere. Namely, we putted more focus on the size, morphology, and the stability of the particles as well as their behavior after redispersion. We are well aware that our study requires some additional studies, among others also optimization of the spray drying process parameters, as also claimed by the Reviewer in the following points, and focusing not only on particle size but also on other process outcomes that are important from an industrial point of view (such as process yield).

Since we did not weigh our product, we cannot give the exact mass of the collected particles. However, we can say that during the optimization phase of the parameters of the spray drying process, we noticed a much larger amount of the sample in the cyclone walls compared to the collection vessel. This is also one of the reasons why we decided to collect and evaluate the samples separately. However, we know that with further process optimization it would be possible to get more samples in the collection vessel, but as mentioned we preferred to focus on our main objective, which was more of our interest. In future studies, we will definitely extend our current results to a much larger extent, taking into account all the points highlighted by the Reviewer.

Point 9: Line 267 ff: in terms of scale-up: how should a powder collection of this formulation should work in an industrial scale? Please comment on this.

Response: We are aware that collecting particles from parts other than the collection vessel is most likely not possible on an industrial scale, since the equipment is of a very different size and the samples are usually taken only from the collection vessels.

However, many authors involved in spray drying at the research level have reported in previous studies that they collect particles from different parts of the spray drying chamber (the collection vessel, the cyclone walls, and even the drying chamber) and combine them into a common sample (DOI: 10.3390/pharmaceutics13081177), which encouraged us to proceed in the same way.

Point 10: Chapter 1.1 The size of the dry particles after spray-drying: DoE approaches are interesting but, in this case, it becomes clear that they cannot describe such a complex system. 48 experiments were proposed by the DoE-program and performed but the discussions starting from Line 303 showed that this DoE does not result in a clear statement, e.g. in regards of temperature influence. In the end, the formulation resulting in the smallest particle sizes E48 was chosen but without clearly understanding the influences and further optimizing the parameters based on the results of the DoE. This chapter has to be revised and restructured and it has to be considered if this DoE approach could really serve as a basis for this parameter optimization.

Response: We thank the reviewer for the comment and agree that the description of the model outcome proposed by DoE was too complicated in the manuscript, explaining every single correlation of the parameters, which probably only confuses the readers. We have rewritten this particular chapter, hopefully in a more reader-friendly way, by focusing on one parameter and its proposed values at a time. We very much hope that the Reviewer will find this acceptable.

However, we are aware that after performing this particular DoE, we obtained process parameters that are not completely optimal. Moreover, our results differ from some previously published results. But spray drying is known to be a very complex process where many different processes and formulation parameters can affect the results. This means that different formulations with the same process parameters can lead to very different particle properties (size, morphology), as reported previously (doi:10.1016/j.ejps.2018.10.026). For this reason, each individual formulation should be optimized separately to achieve the desired result, which was the research track we have followed.

Certainly, the spray drying process could be further improved, e.g., by additional experimental design. However, we believe that we obtained a certain range of process parameters that, according to the final optimized protocol (E48), gave us the result we were aiming for our main objective, namely the smallest particles potentially suitable for our purpose.

We sincerely hope that by using the final process parameters suggested by DoE's model, we have at least provided some sort of proof of concept in our study, and we very much hope that the Reviewer will find this acceptable.

Point 11:    Figure 3: please add lines for the numbers on the y-axis.

Response: The lines have been added, and the figure in the manuscript has been replaced accordingly.

Point 12: Figure 3: clear differences in the encapsulation efficiency (without AA and with AA) were shown in regards to the location of the sampling. What is the reason for this? Please discuss.

Response: Like the Reviewer, we were surprised by the result. However, looking more closely at the spray drying process, we observed that the temperature in the cyclone walls was generally slightly higher than that in the collection vessel, possibly affecting the stability of the drug during the process and leading to lower encapsulation efficiency.

However, we are also aware that sampling at the cyclone walls is not optimal because, as mentioned earlier, it is most likely not suitable for the subsequent scale-up studies. In this sense, looking at our study, we can say that our results confirm that the collection vessel is the most suitable place for sampling, which also gives us the best results.

We inserted the relevant sentence in the manuscript to explain the phenomena, namely: “The lower encapsulation efficiency of all three particle types deposited on the cyclone walls could possibly be due to the higher temperature we observed in the cyclone compared to the collection vessel, which could affect the stability of the drug during the spray drying process”. We really hope that the reviewer will find this acceptable.

Point 13: Line 414: typo “lover”

Response: We thank the Reviewer for the observation. It was corrected accordingly.

Point 14: Line 438: “both storage conditions”: what storage conditions were used regardless of room temperature? This information is only given in graph 4. Please add this information in the methods-section and in the text.

Response: We have included the information in the method section.

Point 15: Line 434: “Larger particles most likely indicate a higher amount of antiadhesive in the particles and thus around the drug, which may protect it from degradation, as shown by our results”. Why is there a higher amount of antiadhesive in larger particles? The ratio of the drug to antiadhesive should be the same in the whole formulation.

Response: We agree with the Reviewer and apologize for the error. Definitely, if the emulsion is stable and the process is consistent, the ratio between the drug and the antiadhesive should be consistent as well. And since we have shown that our emulsions are consistent and stable, this should also apply to our final product.

We have deleted the incorrect statement from the text and reworded the corresponding section. We very much hope that the Reviewer will find this acceptable.

Point 16: Line 439: it is not clear why the amorphous solid state of the particles has t result in a lower chemical stability of the drug after only one month. Please comment on this.

Response: Amorphous substances lack the three-dimensional, long-range order that exists in crystalline materials. The amorphous state is a high-energy state that exhibits enhanced solubility and dissolution rate, and thus increased bioavailability (DOI:10.1016/s0169-409x(01)00098-9). Moreover, the physical and chemical stability of amorphous solids is lower than that of the corresponding crystalline form, and thermodynamically, the amorphous form tends to convert to the more stable crystalline form during processing, storage, or even administration (https://doi.org/10.1016/j.ejpb.2008.07.010). In addition, amorphous solids exhibit higher molecular mobility than the corresponding crystalline forms. The molecular mobility of the amorphous state in relation to chemical reactivity has been documented, with increasing molecular mobility leading to increased chemical degradation of amorphous forms (https://doi.org/10.1002/jps.20926).

We agree that the statement highlighted by the Reviewer lacks explanation and clarity, which has now been added to the text. Also, the DSC thermograms with some additional explanations have been added as supplementary material to provide more clarity to the reader.

We very much hope that the reviewer will find this acceptable.

Point 17: Figure 4: why was the drug content so low for samples with lactose, storage for 1 month at 40 °C?  

Response: Like the reviewer, we are surprised by this result, for which we have found no reasonable explanation. However, we are aware that different types of interactions can occur between different substances and that materials behave differently when combined. This means that the rearrangement of simvastatin around lactose could potentially be different than when we include NCC or incorporate it into CaSi alone. This could potentially lead to oxygen or water penetrating the drug more easily and degrading it more rapidly under more extreme environmental conditions. However, since we saw that after the addition of AA to the formulation, the stability of SIM in the lactose particles increased significantly after 1 month at 40 °C, this means that the antioxidant preserved the drug from oxidative degradation and thus from its chemical stability.

In order to investigate the rearrangement of the substances in the particles, some additional analytical techniques such as Raman spectroscopy should be applied, which should definitely be considered in subsequent studies.

In addition, during storage, there could also be a very small transition of the solid state of simvastatin from the amorphous to the crystalline form (https://doi.org/10.1016/j.ejpb.2008.07.010), which was not detectable by our DSC method, but for which X-ray powder diffraction (XRPD) should be used. And these small changes in the solid state might have affected the chemical stability of the drug, which was different for the different antiadhesives used (namely, the transition could possibly occur for NCC/lactose and CaSi particles, resulting in higher chemical stability after 1 month of storage).

As mentioned earlier, there are several additional tests that can be performed to get a complete picture of what is going on inside the particles that should definitely be investigated in depth in the future.

Point 18: Line 506: please rephrase this sentence for better clarification.

Response: We have rephrased the following sentence: “The particle size measured after redispersion shown in Figure 6 were demonstrated to have a great similarity with the particle size evident from the images of SEM (Figure 2a), indicating the presence of clusters in the medium and that the clusters were neither broken up nor enlarged during the redispersion itself,” for: “The particle size measured after redispersion in Figure 6 shows a close resemblance to the particle size evident from the images of SEM (Figure 2a). This indicates the presence of particle clusters in the medium and that the clusters were neither broken up nor enlarged during redispersion itself.”

We sincerely hope this provides more clarity for readers.

Point 19: Lines 609 ff: this DoE does not allow a detailed discussion of optimized parameters. See comment above.

Response: We have rephrased the sentence from: “Using DoE, we have developed a model proposing optimized parameters for the spray drying process that yields the smallest possible particles whose morphology differs depending on the antiadhesive used,” to: “Using DoE, we have developed a model that proposes parameters for the spray drying process that led us to the smallest particles, whose morphology differs depending on the antiadhesive used.”

Thus, we have excluded the word “optimized” because we agree that the process could be further optimized. We very much hope that the Reviewer will find this acceptable.

Point 20: Line 626: The storage times were not long enough to enable a statement regarding chemical and physical stability. In this case it has to be mentioned in the conclusion that the study was only performed over 1 month.

Response: We thank the Reviewer for the comment and fully agree with the statement, so we have rewritten the conclusion according to the Reviewer's suggestion.

We have changed it to: “After 1 month of storage, the NCC/lactose particles prepared in this manner represent a dry suspension with satisfactory chemical stability of the drug and exhibit adequate size and physical stability after their subsequent redispersion in a suitable printable blank ink medium. With following long-term stability testing, the NCC/lactose dry particles could possibly have the potential to be used in 2D printing technology as a pre-formulation for the production of printable inks ex tempore”.

We very much hope that the Reviewer will find this acceptable.

Point 21: There are quite a few typos in this text. Please check for spelling, a few are mentioned in the comments.

We thank the Reviewer for the comment. We have revised the spelling throughout the manuscript.

Reviewer 3 Report

The paper entitled "Development of simvastatin-loaded particles using spray drying method for ex tempore preparation of cartridges for 2D printing technology" is very interesting, systematic, and clearly written. The data contained in it show the importance of the correct choice of the parameters of the spray drying process in order to obtain dry particles of the right size. Also, the right choice of excipients (antiadhesives and antioxidants) in the emulsion formulation is important to achieve acceptable physical stability of the dry particles as well as chemical stability of the active ingredient in these particles. The ability to easily redisperse dry particles in printable ink media offers the possibility of easy ex tempore production of dosage forms for individual patient needs in a community or hospital pharmacy using the 2D printing technology.

Minor abbreviations, mostly technical in nature, have been included in the text.

Author Response

CORRECTIONS MADE TO MANUSCRIPT ID: pharmaceutics- 2562134

Title: Development of simvastatin-loaded particles using spray drying method for ex tempore preparation of cartridges for 2D printing technology

We would like to thank the Reviewer for the comments and detailed evaluation of our manuscript. We have thus revised and improved the manuscript according to the comments.

RESPONSE to Reviewer 3 Comment

The paper entitled "Development of simvastatin-loaded particles using spray drying method for ex tempore preparation of cartridges for 2D printing technology" is very interesting, systematic, and clearly written. The data contained in it show the importance of the correct choice of the parameters of the spray drying process in order to obtain dry particles of the right size. Also, the right choice of excipients (antiadhesives and antioxidants) in the emulsion formulation is important to achieve acceptable physical stability of the dry particles as well as chemical stability of the active ingredient in these particles. The ability to easily redisperse dry particles in printable ink media offers the possibility of easy ex tempore production of dosage forms for individual patient needs in a community or hospital pharmacy using the 2D printing technology.

Point 1: Minor abbreviations, mostly technical in nature, have been included in the text.

Response: We thank the Reviewer for the positive feedback and for the corrections that will certainly lead to an improved manuscript.

We have added and completed all the points highlighted by the Reviewer in the attached manuscript. The corrections in the revised manuscript are highlighted in yellow.

Round 2

Reviewer 2 Report

Revised manuscript can be accepted. Thank you for your interesting study.